# Modified activity-based anorexia paradigm dampens chronic food restriction-induced hyperadiponectinemia in adolescent female mice

Toru Kuriyama[1], Yusuke Murata[1]*, Reika Ohtani[1], Rei Yahara[1☉], Soichiro Nakashima[1☉], Masayoshi Mori[1], Kenji Ohe[1‡], Kazunori Mine[2], Munechika Enjoji[1‡]

1 Department of Pharmacotherapeutics, Faculty of Pharmaceutical Sciences, Fukuoka University, Fukuoka, Japan, 2 Faculty of Neurology and Psychiatry, BOOCS Clinic Fukuoka, Fukuoka, Japan

☉ These authors contributed equally to this work.
‡ KO and ME also contributed equally to this work.
* ymurata@fukuoka-u.ac.jp

**Data Availability Statement:** All relevant data are within the paper.

## Abstract

Anorexia nervosa (AN) is a chronic, life-threatening disease with mental and physical components that include excessive weight loss, persistent food restriction, and altered body image. It is sometimes accompanied by hyperactivity, day-night reversal, and amenorrhea. No medications have been approved specific to the treatment of AN, partially due to its unclear etiopathogenesis. Because adiponectin is an appetite-regulating cytokine released by adipose tissue, we hypothesized that it could be useful as a specific biomarker that reflects the disease state of AN, so we developed a modified AN mouse model to test this hypothesis. Twenty-eight 3-week-old female C57BL/6J mice were randomly assigned to the following groups: 1) no intervention; 2) running wheel access; 3) food restriction (FR); and 4) activity-based anorexia (ABA) that included running wheel access plus FR. After a 10-day cage adaptation period, the mice of the FR and ABA groups were given 40% of their baseline food intake until 30% weight reduction (acute FR), then the body weight was maintained for 2.5 weeks (chronic FR). Running wheel activity and the incidence of the estrous cycle were assessed. Spontaneous food restriction and the plasma adiponectin level were evaluated at the end of the acute and chronic FR phases. An increase in running wheel activity was found in the light phase, and amenorrhea was found solely in the ABA group, which indicates that this is a good model of AN. This group showed a slight decrease in spontaneous food intake accompanied with an attenuated level of normally induced plasma adiponectin at the end of the chronic FR phase. These results indicate that the plasma adiponectin level may be a useful candidate biomarker for the status or stage of AN.

## Introduction

Anorexia nervosa (AN) is an eating disorder characterized by excessive weight loss, persistent food restriction, and altered body image [1]. Among all psychiatric disorders, AN has the

**Funding:** The author(s) received no specific funding for this work.

**Competing interests:** The authors have declared that no competing interests exist.

highest mortality because of somatic complications associated with undernutrition and suicide [2, 3]. Half of all inpatients with AN experience relapse within a year, and approximately 20% of patients remain chronically ill [4]. The lifetime prevalence of AN is estimated to be 1 to 4% [5] and it is of importance that the incidence of eating disorders increased among young people during the COVID-19 pandemic [6]. Unfortunately, no specific medications have been approved for the treatment of AN [7], partially due to the fact that the etiopathogenesis of AN remains unclear. Thus, valid animal models of AN must be prepared to clarify the pathogenesis and develop therapeutic agents for AN.

When developing experimental animal models of AN, most studies have adopted an activity-based anorexia (ABA) paradigm that generally consists of wheel running access and time-restricted feeding [8]. The rodents exposed to an ABA paradigm exhibit numerous clinical characteristics of AN, such as severe weight loss, starvation-induced hyperactivity, cessation of the estrous cycle, and heightened vulnerability during puberty [9]. However, the mortality rate of animals subjected to the conventional ABA paradigm has been high after chronic starvation due to the rodents having continued to run even during the feeding time [10, 11]. Thus, based on the clinical fact that AN tends to lead to a chronic disease course, a modified model that avoids mortality by mice under the chronic starvation needs to be adopted. Méquinion et al. [12] modified the initial ABA paradigm of a chronic mouse model based on quantitative food restriction associated with voluntary exercise in a running wheel: the "Food Restriction and Wheel (FRW)" model. Frintrop et al. [13] established a modified FRW rat model with daily adjustment of food consumption to achieve a target weight reduction that prevents over-starvation. Of note, remarkable changes were seen in 4-week-old female rats that had a 25% reduction of body weight and maintained this starvation level for an additional 2 weeks compared with 8-week-old animals [13]. Because the onset of AN occurs mostly during adolescence [14], the chronic model of Frintrop et al. is suitable for a preclinical approach to elucidate the complex etiology of AN, but their model was only validated using female rats. The first purpose of the present study was to validate a modified FRW model that uses adolescent female mice to mimic the symptomatic features of clinical AN, such as hyperactivity, amenorrhea, and spontaneous food restriction. Only female mice were used because of a higher incidence of AN in women and girls than in men and boys (sex ratios of approximately to 10:1 to 15:1 [7]).

AN is associated with multiple, profound endocrine disturbances that can be adaptive, reactive to chronic starvation, or etiologic [15, 16]. Many hormones related to glucose metabolism, the hypothalamic-pituitary-adrenal axis, adipokines, and appetite-regulating hormones have the potential to serve as useful clinical biomarkers of the severity and treatment response of AN patients [16]. Although adiponectin is an appetite-regulating cytokine secreted from adipose tissue, the blood adiponectin level is negatively correlated with body mass index (BMI) and body fat mass [17]. In line with this, most studies and a recent meta-analysis reported that patients with AN exhibited hyperadiponectinemia, which is normalized during refeeding [18, 19]. However, elevated blood adiponectin has been observed not only in AN patients, but also in healthy underweight young people (BMI < 18.5) [20]. Our second aim was to investigate the effects of acute and chronic food restriction on the plasma adiponectin levels of experimental groups based on the presence or absence of wheel running and target weight loss.

In light of the above, we developed a modified mouse model for this study that would allow us to determine if circulating adiponectin can be used as a specific biomarker that reflects the disease state of AN and to elucidate how the effect of weight loss on the blood adiponectin level differs between 1) weight control intervention (dietary restriction and experimental AN) and 2) the acute and chronic phases of AN.

## Materials and methods

### Animals and housing conditions

A total of twenty-eight 3-week-old female C57BL/6J mice were purchased from Charles River Laboratories Japan, Inc. (Yokohama, Japan), with each individually housed in a polypropylene cage from arrival. All were kept under the following conditions: 23 ± 2 ˚C, absolute humidity 60 ± 2%, and a 12/12-h light/dark cycle (7:00–19:00: light period, 19:00–7:00: dark period). The animals were provided access to standard food and water to be consumed ad libitum for ten days after delivery. Animal experiments were conducted in accordance with the ethical guidelines for animal experiments by the Experimental Animal Care and Use Committee of Fukuoka University (approval number: 2015119 and 2112110; date of approval: 22/3/2021 and 13/12/2021), which follows universal principles of laboratory animal care.

### Food restriction

A consecutive three and half weeks of food restriction (FR) was conducted according to the description of Frintrop et al. [13], with minor modification. Briefly, the FR sessions were separated into acclimatization, acute, and chronic phases. In the acclimatization phase, the average daily food intake for 10 days from arrival was calculated for each mouse in order to determine the daily consumption of food to be given in later phases. At day 10, the body weight was measured and used as baseline. In the acute FR phase, from day 11, animals received only a fixed amount of food (40% of the daily food intake). When the mice reached a 30% weight reduction from baseline, the acute FR phase ended. On average, the acute FR phase continued for 6.9 ± 0.4 days. The chronic FR phase (weight holding phase) was then started, in which the mice were given an amount of food that had initially been increased to 50% of the daily food intake during the acclimatization phase. To maintain the 30% weight loss from baseline, the amount of food was adjusted daily. If the weight deviated by more than 2.5% from the target weight, the amount of food the next day was increased or decreased in increments of 5%. The chronic FR phase was continued until day 35. Pre-weighed food pellets were given from 15:00 to 17:00 during the acute and chronic FR phases. The current study obtained approval of the University Ethics Committee, even though the 60% FR necessary to achieve a 30% body weight loss seems to be very severe when the acceptable weight loss limit is usually considered 20% for adult mice.

### Running wheel activity

At delivery, a portable running wheel (LCW-M4, Melquest, Japan) was placed on the floor of each housing cage. The wheel, 155 mm in diameter, was linked to a rotation data recording system (CIF-4/Actmaster®, Melquest). The number of rotations of each wheel in 1-min bins was continuously monitored. The running wheel activity was calculated separately for the light (7:00–19:00) and dark phases (19:00–7:00).

### Estrous cycle

Estrous cycle was determined according to previous descriptions [21–23]. Briefly, the trunk of the mouse was gently pressed to immobilize it while the tail was lifted upward. The end of a sterile 200 μL-pipette tip filled with double distilled water (ddw, 50 μL) was placed at the opening of the vaginal canal, then the ddw was gently expelled into the canal and the mixture of water and internal fluid was withdrawn into the tip. Using the same tip, expulsion and aspiration procedures were repeated 4–5 times, then the fluid was dropped on glass slide. The wet smear was allowed to completely dry at room temperature for at least 1-h. The slides were

incubated in 0.1 w/v% crystal violet solution (031–04852, FUJIFILM Wako Pure Chemicals, Japan) for 1 min, then washed twice in ddw, air-dried, and coverslipped with Entellan™ new (1.07961, Merck Millipore, Japan). Based on the cell morphology under a microscope with 40× magnification, the stage of the estrous cycle was identified as follows: proestrous, estrous, metestrous, or diestrous. Because the estrous cycle of a healthy mouse takes 4 days [22], the incidence of the estrous cycle was determined by vaginal smear cell morphology in 4-day blocks that met any of the following conditions: transition from proestrous to estrous, estrous, or transition from estrous to proestrous.

## Experimental protocol

At arrival, all mice were randomly assigned to one of the following four experimental groups: 1) an *ad libitum* (AL) group (N = 6) was housed in a standard cage without a running wheel and allowed to consume food pellets *ad libitum* throughout the experiment; 2) a wheel running (WR) group (N = 8) was housed in a standard cage with a running wheel and allowed to consume food pellets ad libitum throughout the experiment; 3) a food restriction (FR) group (N = 6) was housed in a standard cage without a running wheel and allowed to consume food pellets *ad libitum* until day 10. From days 11 to 35, the food intake was restricted in order to achieve and maintain a 30% weight loss from baseline body weight; and 4) an activity-based anorexia (ABA) group was housed in a standard cage with a running wheel and allowed to consume food pellets ad libitum until day 10. From days 11 to 35, the food intake was restricted in order to achieve and maintain a 30% weight loss from baseline body weight. The body weight and cumulative food intake were measured daily throughout the experiment. Because the body weight fluctuations were stabilized, weighing was done every other day from day 28 to 35. The wet smear in all groups was collected daily from day 12, the day after the acute FR phase started. At the end of the experiment, all mice were sacrificed as described below. The experimental schedule for each treatment is shown in Fig 1.

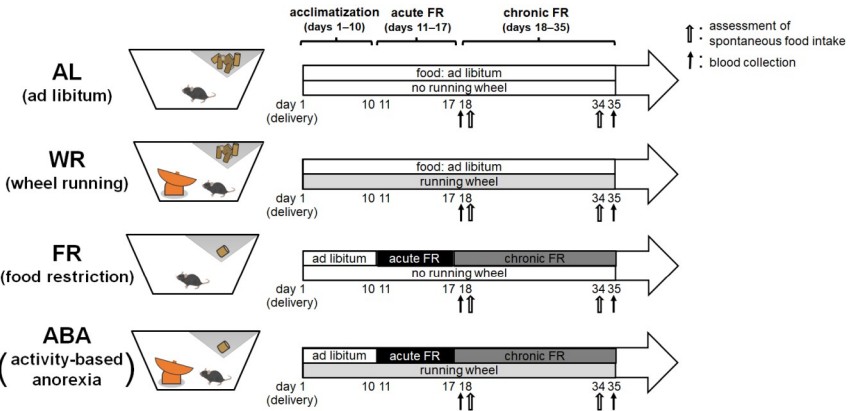

**Fig 1. Experimental design.** 1) *ad libitum* (AL) group (N = 6) animals were not subjected to any intervention from arrival. 2) wheel running (WR) group (N = 8) animals were subjected to running wheel access from arrival to the end of the experiment. 3) food restriction (FR) group (N = 6) animals were subjected to 10-days adaptation, then received 40% of their baseline food intake until a 30% weight reduction was reached (acute FR). The body weight was then maintained for 2.5 weeks (chronic FR). 4) activity-based anorexia (ABA) group (N = 8) animals were subjected to running wheel access from arrival and weight reduction intervention similar to the FR group from day 11. Body weight, food intake, and running wheel activity (only in the WR and ABA groups) were measured daily from day 1 to 28 and every other day from day 28 to 34. The estrous cycle was measured in 4-day blocks from the start of the acute FR session. Spontaneous food intake was estimated at days 18 and 34. Blood collection was done on days 18 (via the tail vein) and 35 (via cardiac puncture).

## Spontaneous food restriction

At days 18 and 34, spontaneous food restriction, a characteristic feature of human AN, was assessed using a protocol based on our preliminary data (original protocol referring to the study by Suyama et al. [24]). Briefly, all mice were allowed to eat food pellets freely for 90 min, and the amount of food consumed was measured three times, every 30 min. Of note, the AL and WR groups were not subjected to this assessment to avoid the adverse effects of food deprivation. Only a fixed amount of food necessary to maintain the target weight was given to the FR and ABA groups. This process started between 15:00 and 16:00.

## Plasma collection and assessment of the adiponectin level

At day 18, the mice were taken from their home cage and placed into a gentle-restraint cage (KN-325-C-1, Natsume Seisakusho Co., Ltd., Japan). The lateral tail vein was immediately slightly incised using a razor blade, and an approximately 5 µL of blood sample was collected into a heparinized glass capillary tube (9100275, EM MYSTAR hematocrit capillary; As one, Japan). For the FR and ABA groups, this process was conducted before the assessment of spontaneous food restriction. At day 35, the end of the chronic FR phase, the mice were anesthetized by intraperitoneal injection of a mixture of medetomidine (0.3 mg/kg), midazolam (4 mg/kg) and butorphanol (5 mg/kg) at a dose determined according to the report of Kawai et al. [25]. Then, the heart was exposed through an incision in the chest and approximately 500 µL of the trunk blood was collected into a 1 mL syringe that contained sufficient heparin to wet the inner wall and fill the dead space using cardiac puncture. The mice were then decapitated.

The blood samples were centrifuged at 3,000 rpm for 15 min at 4 ˚C to separate the plasma. The adiponectin level in plasma was assayed using a commercially available ELISA kit according to the manufacturer's instructions (47579900, Mouse/Rat Adiponectin ELISA-OY, Oriental Yeast Co. Ltd., Japan).

## Statistical analysis

The Shapiro-Wilk and Levene's tests were used with R software version 4.2.1 to test the normality of the distribution of the data and the homogeneity of the variance [26]. All statistical analyses for measures were done using StatView software Ver.5 (HULINKS, Tokyo, Japan). When the measures were not distributed normally, a non-parametric test was adopted. Two-way repeated analysis of variance (ANOVA) was performed to evaluate the significance of differences between the FR and ABA groups in the amount of food consumed at day 34 of the spontaneous food restriction period, followed by Bonferroni/Dunn post hoc analysis. Two-way factorial ANOVA was done to evaluate the significance of differences in the plasma adiponectin levels of the four experimental groups at day 18, followed by Bonferroni/Dunn post hoc analysis. The Kruskal-Wallis test was performed to evaluate the effect of FR and WR on the averaged data on body weight and daily food intake from day 1 to 10 (acclimatization phase), day 11 to 17 (acute FR phase), and day 18 to 34 (chronic FR phase) and on the plasma adiponectin levels at day 35, followed by post hoc Mann-Whitney $U$ tests with Bonferroni correction to adjust for multiple comparisons ($\alpha = 0.05/6 = 0.0083$). The Friedman test was done to test for differences for food intake between the FR and ABA groups at day 18 of the spontaneous food restriction, followed by post hoc analysis using the Mann-Whitney $U$ test with Bonferroni correction for multiple intergroup comparisons ($\alpha = 0.05/3 = 0.0166$) and the Wilcoxon's signed-rank test with Bonferroni correction ($\alpha = 0.05/6 = 0.0083$). The Mann-Whitney $U$ test was used to evaluate differences between the WR and ABA groups in daily wheel running activity in the light and dark phases from day 1 to 10, day 11 to 17, and day 18 to 34. The Chi-

squared test was used to analyze the data on the incidence of estrous phase within the 4-day blocks. All data are presented as mean ± S.E.M. A *p* value of less than 0.05 was considered statistically significant.

# Results

## Body weight and food intake

Fig 2 represents change over time in body weight and cumulative food intake throughout the experiment. The body weight and cumulative food intake were markedly lower in the food-restricted groups (FR and ABA) than in the freely-fed groups (AL and WR; Fig 2). The averaged data for body weight and food intake during the three experimental phases are shown in Fig 3. For average body weight, the Kruskal-Wallis test revealed significant differences among the experimental groups during the acute ($H = 20.9$, $p < 0.001$; Fig 3A *medium*) and chronic FR phases ($H = 20.4$, $p < 0.001$; Fig 3A *right panel*), with a non-significant trend found in the acclimatization phase ($H = 6.57$, $p = 0.087$; Fig 3A *left panel*). *Post-hoc* analysis further showed that the means of body weight of the FR and ABA groups were significantly lower than those of the AL and WR groups in the acute ($p < 0.00833$, AL v.s. FR, AL v.s. ABA, and WR v.s. FR; $p < 0.00167$, WR v.s. ABA) and chronic FR phases ($p < 0.00833$, AL v.s. FR, AL v.s. ABA, WR v.s. FR, and WR v.s. ABA). Kruskal-Wallis test for the averaged food intake revealed significant differences among the experimental groups during the acclimatization ($H = 11.2$, $p < 0.05$; Fig 3B *left panel*), acute ($H = 21.0$, $p < 0.001$; Fig 3B *medium*), and chronic FR phases ($H = 22.5$, $p < 0.001$; Fig 3B *right panel*). *Post-hoc* analysis further showed that the food intake of the ABA group was significantly lower than that of the AL group ($p < 0.00833$). The mean food intakes of the FR and ABA groups were significantly lower than those of the AL and WR groups in the acute and chronic FR phases ($p < 0.00833$, AL v.s. FR, AL v.s. ABA, and WR v.s. FR; $p < 0.00167$, WR v.s. ABA in each phase). Of note, in the chronic FR phase, the food intake of the WR group was significantly higher than that of the AL group ($p < 0.00833$).

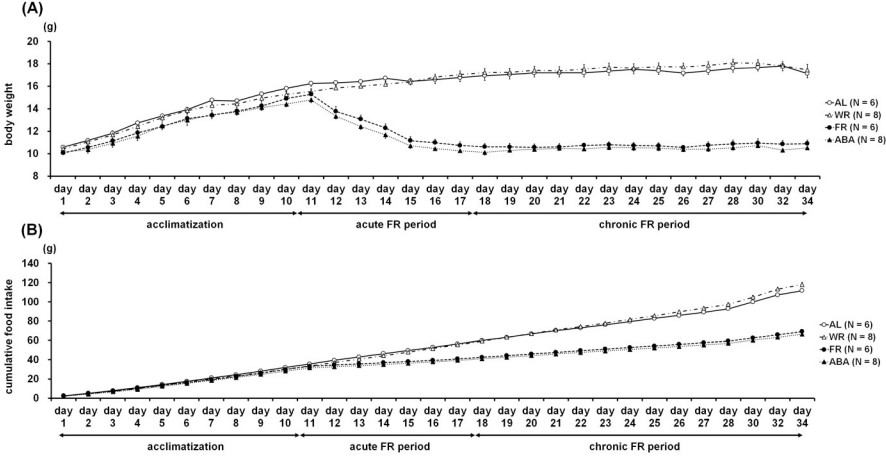

**Fig 2. Time course of body weight (A) and cumulative food intake (B).** In the FR and ABA groups, 30% of the body weight was lost on day 10 due to food being given at only 40% the amount of the daily food intake in the adaptation period. After achievement of a 30% weight loss, the amount of food given was increased to 50% in order to maintain the target weight.

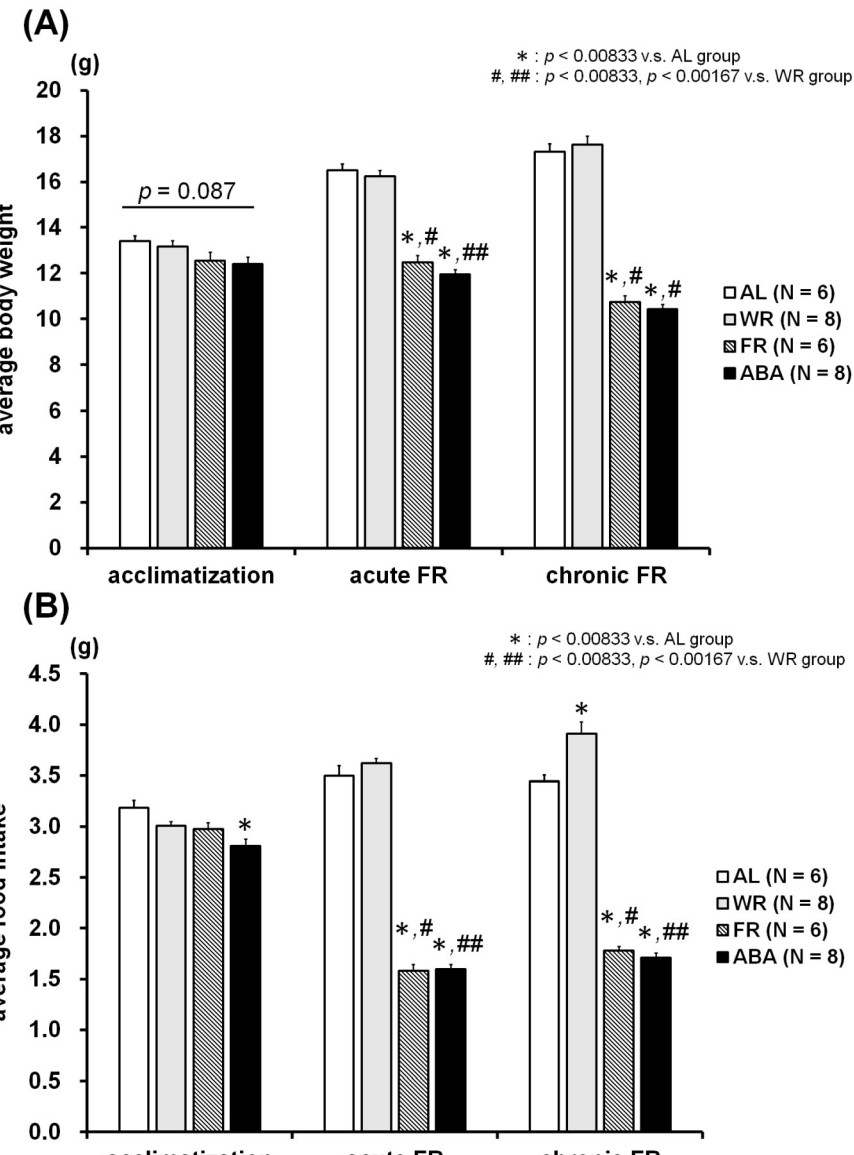

**Fig 3. Average body weight (A) and daily food intake (B) during three experimental sessions: Acclimatization (days 1 to 10), acute FR (days 11 to 17), and chronic FR (days 18 to 34).** In the FR and ABA groups, it is not surprising that the body weight and daily food intake were significantly lower than in the other two groups.

## Running wheel activity

Fig 4 represents the change over time in running wheel activity of the WR and ABA groups. All mice were actively spinning the wheel during the dark phase and resting in the light phase; however, the rotations of the ABA group in the light phase were considerably increased from day 14 while those in dark phase continuously decreased until the end of experiment. The averaged data for running wheel activity during the three experimental phases is shown in Fig 5. The Mann-Whitney $U$ test revealed significant differences in average activity between the WR and ABA groups during the acute (light phase, $Z = -3.36$, $p < 0.001$; Fig 5A *medium*; dark phase, $Z = -2.10$, $p < 0.05$; Fig 5B *medium*) and chronic FR phases (light phase, $Z = -3.15$, $p < 0.01$; Fig 5A *right panel*; dark phase, $Z = -3.36$, $p < 0.001$; Fig 5B *right panel*).

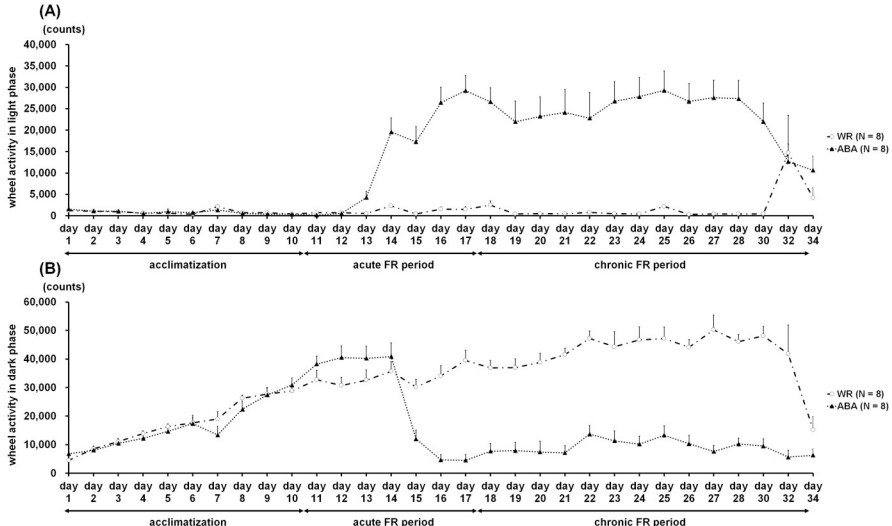

**Fig 4. Time course of running wheel activity in the light phase (A) and the dark phase (B).** In the ABA groups, a considerable increase in activity in the light phase and the relative decrease in the dark phase were found from day 14.

## Estrous cycle

Fig 6 and Table 1 represent the four-day block data for the incidence of the estrous cycle from the acute FR phase until the end of the experiment. Chi-squared test revealed significant differences at block 2 (the latter half of the acute FR phase, $\chi^2 = 15.7$, $p < 0.01$) and at blocks 3 to 5 (chronic FR phase; block 3, $\chi^2 = 24.5$, $p < 0.001$; block 4, $\chi^2 = 15.7$, $p < 0.01$; block 5, $\chi^2 = 18.7$, $p < 0.001$).

## Spontaneous food restriction

Fig 7 represents the spontaneous food intake, calculated every 30 min for 90 min, of the FR and ABA groups at days 18 and 34. The Friedman test used for intra-group comparisons showed significant differences in spontaneous food intake at day 18 (WR group, $\chi^2 = 18.0$, $p < 0.001$; ABA group, $\chi^2 = 23.4$, $p < 0.001$, Fig 7A). Pairwise comparisons (Wilcoxon rank order tests) indicated that food intake increased gradually over the 90 min. Inter-group analysis indicated no significant differences between these two groups. In contrast, two-way repeated ANOVA revealed a significant major effect of time ($F_{1, 36} = 2070$, $p < 0.001$) and a significant group × time interaction ($F_{3, 36} = 3.58$, $p < 0.05$, Fig 7B). *Post-hoc* analysis further showed a non-significant trend toward a lower intake in the ABA group at 90 min ($t_{1, 12} = -2.06$, $p = 0.062$).

## Plasma adiponectin levels

Fig 8 represents the plasma adiponectin levels of the four experimental groups at days 18 and 35. Two-way factorial ANOVA revealed a non-significant trend in the difference in plasma adiponectin level at day 18 of the food-restricted and freely-fed groups ($F_{1, 24} = 4.12$, $p = 0.054$; Fig 8 *left panel*). In contrast, the Kruskal-Wallis test revealed a significant difference at day 35 ($H = 19.3$, $p < 0.01$; Fig 8 *right panel*). *Post-hoc* analysis further showed significantly more circulated adiponectin in the FR group than in the AL and WR groups (each $p < 0.00833$). However, the plasma adiponectin level of the ABA group was significantly higher than that of WR

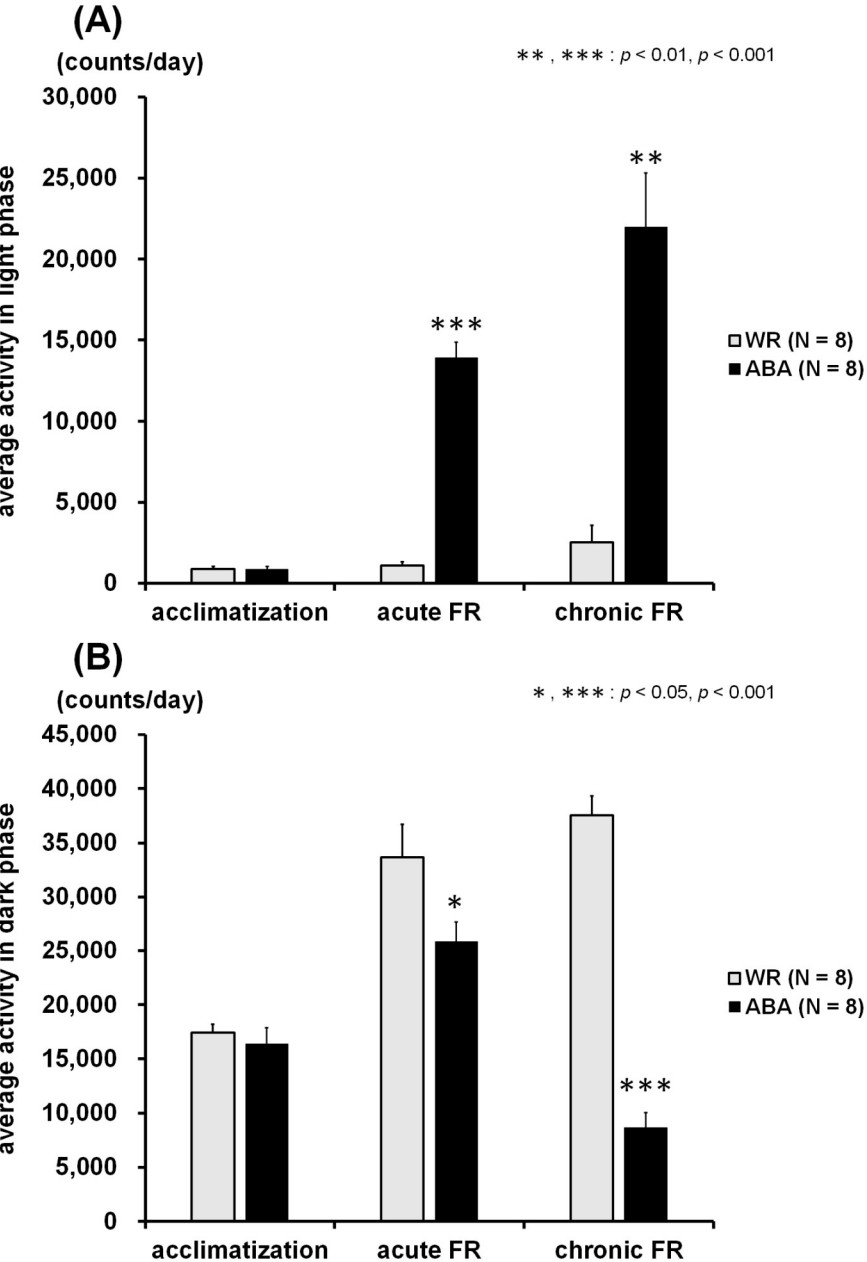

**Fig 5. Average running wheel activity in the light phase (A) and the dark phase (B) during three experimental sessions: Acclimatization (days 1 to 10), acute FR (days 11 to 17), and chronic FR (days 18 to 34).** In the ABA group, the activity in the light phase was significantly higher and the activity in dark phase was significantly lower than that of the WR group during acute and chronic FR sessions.

($p < 0.00833$) but not AL. Of note, a non-significant trend toward a lower adiponectin concentration was found in the ABA group compared to the FR group ($p = 0.01$).

## Discussion

This study was done: 1) to validate our newly-modified ABA paradigm (originally developed by Méquinion et al. [12] and Frintrop et al. [13]) that uses adolescent female mice to model

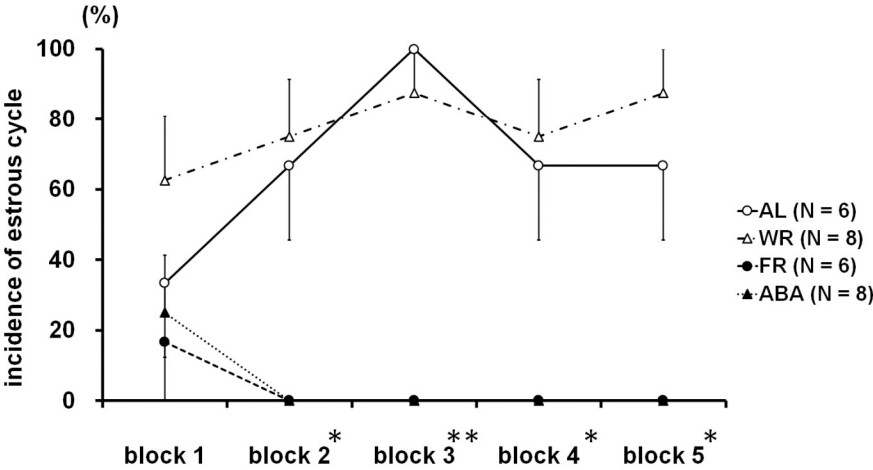

**Fig 6. Time course of incidence of estrous cycle in 4-days blocks from the acute FR session.** In the FR and ABA groups, no estrous cycle was found after the second block.

AN and 2) to investigate the combined effects of target weight loss and wheel running on the plasma adiponectin level during the acute and chronic FR phases.

Our findings showed that post-weaning female mice subjected to this ABA paradigm exhibited hyperactivity in the resting phase, amenorrhea, and a tendency to spontaneously restrict feeding, despite starvation (It should be noted that no mice died under the acute and chronic FR interventions). These characteristic features closely resemble the clinical symptoms of AN, as shown in previous reports [8, 12, 13], which indicates that our ABA paradigm is appropriate for developing an adolescent female AN mouse model. However, several notable issues need to be discussed. First, the mice in the ABA group tended to be lighter and had less appetite than the mice of the other three groups during the acclimatization phase (Fig 3A and 3B *left panel*). This may be due to a difference in the age of the animals at delivery. We ordered and purchased 3-week-old female mice, however, their ages ranged from 15 to 21 days at arrival. Because changes in body weight and appetite are likely to be larger in younger mice, there may have been differences between the groups at the start of the experiment. Future study will be needed to align the age and body weight at delivery of all experimental groups. Second, the dark phase activity of the ABA group did not increase, but rather showed a significant decrease compared to the WR group (Figs 4 and 5). The conventional ABA protocol that consists of restricted food access for 1 to 2 hours/day and running wheel access leads to increased hyperactivity [11]. However, prior studies have demonstrated that caloric restriction and the ABA paradigm can stimulate diurnal activity, which may reflect exaggerated food anticipatory

**Table 1. Comparisons of incidence of estrous cycle of four experimental groups in 4-days blocks.**

| | Incidence of estrous cycle (Yes, %) | | | | |
|---|---|---|---|---|---|
| | block 1 | block 2 | block 3 | block 4 | block 5 |
| AL (N = 6) | 33.3 | 66.7 | 100.0 | 66.7 | 66.7 |
| WR (N = 8) | 62.5 | 75.0 | 87.5 | 75.0 | 87.5 |
| FR (N = 6) | 16.7 | 0.0 | 0.0 | 0.0 | 0.0 |
| ABA (N = 8) | 25.0 | 0.0 | 0.0 | 0.0 | 0.0 |
| $\chi^2$ | 3.86 | 15.7 | 24.5 | 15.7 | 18.7 |
| p value | 0.277 | < 0.01 | < 0.001 | < 0.01 | < 0.001 |

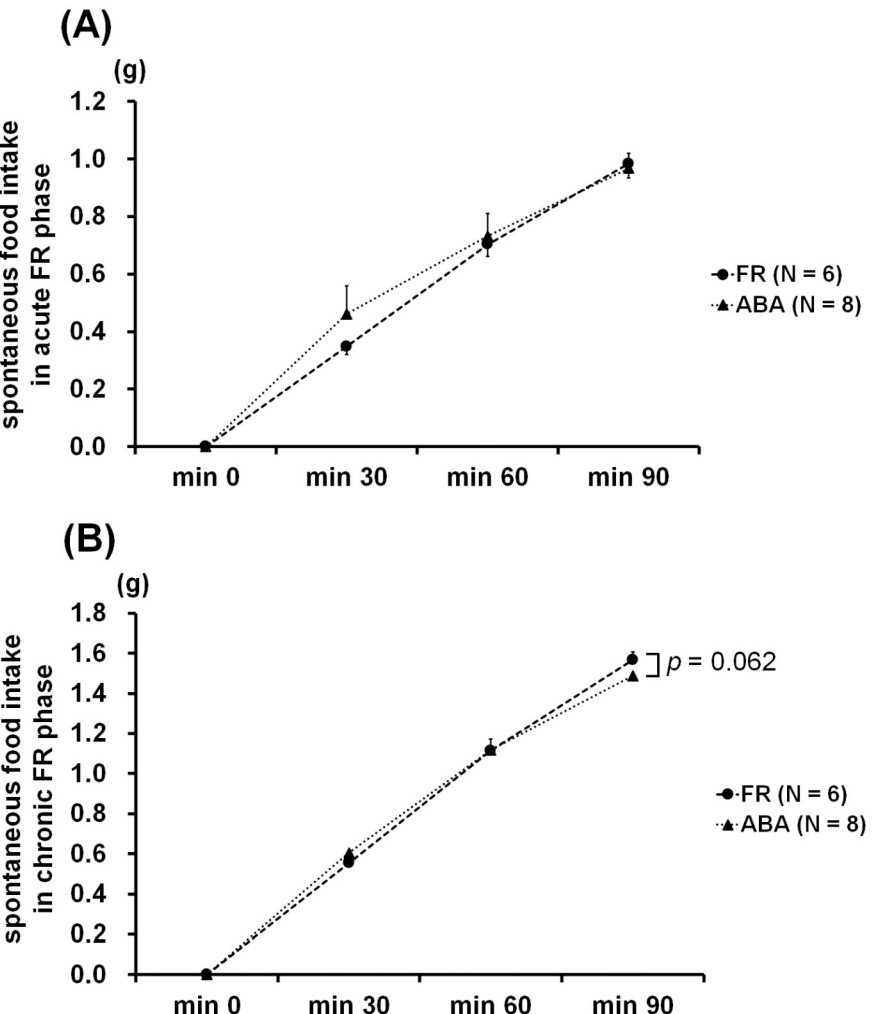

**Fig 7. Time course of spontaneous food intake at days 18 (A) and 34 (B).** At the end of the acute FR session, no significant difference was found between the FR and ABA groups. At the end of the chronic FR session, some spontaneous food restriction was observed in the ABA group.

activity or a shift in circadian activity to promote seeking food [27–30]. Given that disruption of circadian rhythms has been observed in AN [31], our present findings suggest that the ABA paradigm-induced hyperactivity was the result of activation of starvation-induced foraging mechanisms that promote physical activity. In future study, differences in the diurnal/nocturnal locomotor activity in the home cage should be monitored for the FR and ABA groups. Third, the incidence of the estrous cycle varied among the four experimental groups during the first half of the acute FR phase (Fig 6 *leftmost*), which might be attributed to individual differences in sexual maturation. Manzano Nieves et al. [32] investigated the probability of early adolescent female mice being at a given estrous cycle and found that 16.9% of their mice did not have a complete vaginal opening at 35 days. Because the vaginal smear was collected starting from 4.5 weeks in the present study, approximately 20% of the animals in each experimental condition may have been sexually immature. Given that the relative energy deficit in the body associated with weight loss or exercise leads to functional hypothalamic amenorrhea (FHA) [33–35], the decrease in secretion of hypothalamic gonadotrophin-releasing hormone

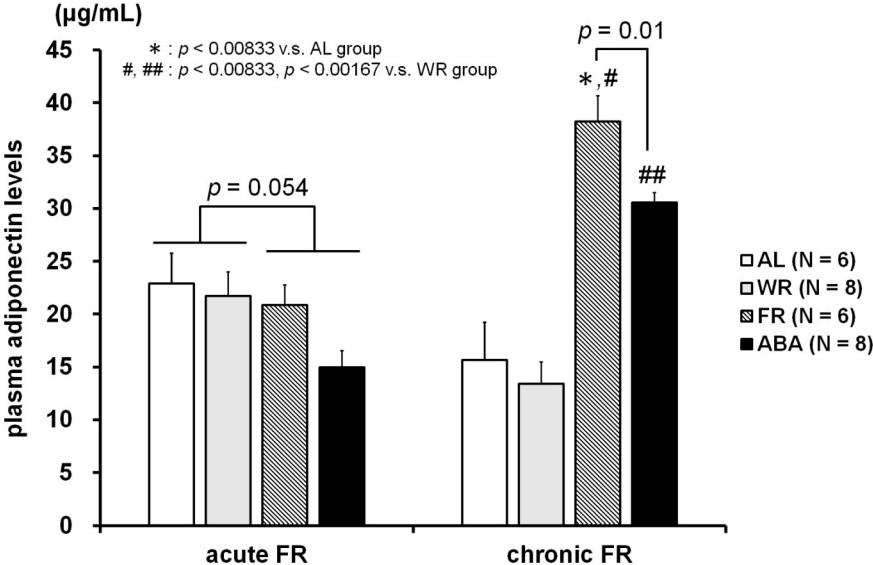

**Fig 8. Plasma adiponectin levels at days 18 and 35.** There was a trend toward differences among the four groups at the end of the acute FR session. At the end of chronic FR phase, the concentration was significantly higher in the FR and ABA groups than in the AL and WR groups. Circulating adiponectin tended to be lower in the ABA group than in the FR group.

may be a better indicator of FHA in ABA-loaded animals than is the cytological identification of the menstrual period. Fourth, we evaluated symptomatic features of AN, such as hyperactivity, amenorrhea, and spontaneous food restriction. However, other characteristics of AN that many studies have assessed; including basal glycaemia, glucose tolerance, bone status, and body composition; were not examined. Future study that evaluates physiological abnormalities will need to be conducted for comparison with previous ABA paradigms, which will emphasize the uniqueness of our model. Lastly, spontaneous food restriction in the ABA group at the end of the chronic FR phase was not statistically significant. Although voluntary dietary restriction is an important pathophysiological feature of AN, few rodent studies with a standard ABA paradigm have found a decrease in food intake in mice. Of importance, in a previous study by Méquinion et al. [12], an important reference for the protocol adopted in the current study, the reduction of spontaneous intake 1 and 3 hours after presentation of a food pellet was found in mice with wheel running and limited food access in which a fixed amount of food was given. The inconsistency may be due to methodological differences, such as the age of the animals, the food presentation time, and/or the duration of intervention. In particular, the most significant difference is how food intake was restricted during the acute and chronic FR phases. Méquinion et al. [12] fed their mice a constant amount of food throughout the experimental period, whereas we adopted a protocol that is able to fine-tune the amount of food given in order to maintain a target weight, thus it is difficult to compare the results of these studies because of the different methodologies. Of interest, because running animals were thirsty and drank a lot before reaching a satiety sensation which decreases their food intake, Boakes and Juraskova [36] demonstrated that use of wet and mashed food suppresses drinking-induced decrement of feeding behavior in an ABA model. Further studies will be necessary after optimizing the experimental conditions.

The present study found a non-significant trend toward a lower plasma adiponectin level in food-restricted mice (FR and ABA groups) than in freely-fed animals (AL and WR groups) at

the end of the acute FR phase, while the plasma adiponectin levels of the FR and ABA groups were significantly higher than those of the AL and WR groups at the end of the chronic FR phase. Adiponectin is an adipokine synthesized and secreted from adipose tissue that regulates glucose and lipid metabolism and that has insulin-sensitizing and anti-atherosclerotic actions [37, 38]. In spite of this being an adipose-derived hormone, it is of interest that the blood adiponectin level is inversely correlated with BMI and body fat mass [17]. Thus, many clinical studies and reviews report that hyperadiponectinemia is frequently observed in AN patients [18, 19, 39–43]. In studies of rodents, the blood adiponectin level was not changed by short-term caloric restriction and/or exercise [44–46]. On the other hand, long-term dietary restriction has been shown to increase the blood adiponectin concentration [46–49]. In the present study, the plasma adiponectin levels were not statistically different among the experimental groups (slightly lower in the ABA group than in the other groups) after 1 week of food restriction and wheel running, while a chronic FR-induced increase in adiponectin was observed after 3.5 weeks intervention. These data are consistent with previous findings, however, it is of interest that less circulating adiponectin was observed in our ABA group than in our FR group at the termination of the experiment. Although little is known about the impact of the ABA paradigm on the blood adiponectin level, Tirelle et al. [50] found no change in the blood adiponectin levels of intact and ABA paradigm-loaded female mice. That study also demonstrated that the plasma leptin concentration of their AL group was comparable to that of their ABA group, which displayed a 25% body weight loss. Because leptin is secreted from adipose tissue, the blood leptin level in ABA-loaded animals has been shown to be reduced by the food restriction-induced decrement of fat mass [12, 51, 52]. It is difficult to understand how there could be a decrease in lean mass and an increase in fat mass in female mice loaded with ABA paradigm, however, it may explain why no alteration in plasma leptin level was found between in AL and ABA group [50]. Despite adiponectin being secreted from adipocytes, the blood adiponectin level has been negatively correlated with body fat mass [17]. Thus, the increased adiposity in female ABA mice of the Tirelle study may have led to the decrease in adiponectin, which should have been high under the poor nutrition state [50]. Although we did not examine fat mass, post-decapitation dissection confirmed that little fatty tissue remained. Our modified protocol with amount-restricted food access is the first study to demonstrate an effect on the plasma adiponectin level of adolescent female mice of achieving a target weight reduction. Of interest, the male ABA mice of the Tirelle study showed higher decreases in fat mass and plasma leptin level than AL mice, but not plasma adiponectin level [50]. Because male and female mice exhibit different responses to the ABA model [53], the ABA-induced increment in adiponectin may be specific to female mice. Indeed, an elevated level of adiponectin has been associated with reduced basal gonadotropin-releasing hormone and luteinizing hormone in mice [54], which results in FHA. Future studies will be needed to clarify whether or not the ABA-induced change in adiponectin is sex-specific.

Adiponectin has been shown to decrease body weight via enhancing energy expenditure and increasing beta-oxidation, but studies have shown that it does not affect the food intake of healthy mice [55, 56]. However, the food intake was increased by adiponectin injection to mice on a 12 hr fasting then 3 hr refeeding schedule [57], which suggests that adiponectin has an appetite-modulating effect based on the state of starvation/satiety. Given that the increase in the blood adiponectin level by long-term caloric restriction implies a compensatory response, we speculate that the dampening of this response in ABA mice contributes to the pathophysiology of AN. In other words, ABA mice have relatively low levels of blood adiponectin, which diminishes the appetite-enhancing effects of adiponectin despite being hungry. In support of our speculation, Buckert et al. [58] investigated temporal changes of the plasma adiponectin levels of AN patients with and without severe symptoms in a clinical intervention

study, only observing hyperadiponectinemia in subjects with mild to moderate severity (BMI 16 or over) at the beginning. However, the plasma adiponectin level of severe AN patients (BMI under 16) increased in increments of BMI up to 16 during the treatment phase, then decreased with further weight gain. The fact that a specific BMI is the threshold for reversing changes in adiponectin concentration in AN patients supports the hypothesis that adiponectin plays an important role in the pathophysiology and treatment of AN [18, 55]. Future studies will be needed to determine the time course of the plasma adiponectin levels of ABA mice, which depend on the magnitude and duration of weight loss.

There are several limitations to the present study. First, the sample size was relatively small. Although we analyzed the data using appropriate statistical methods, a larger study will be necessary to confirm our findings. Second, several endocrinological changes associated with AN have been reported, including insulin, corticosterone, sex steroids, growth hormone, ghrelin, oxytocin, leptin, and others [16]. We analyzed only the plasma adiponectin level, however, it should be noted that this is the first study to examine the longitudinal change of adiponectin under conditions of dietary restriction and exercise. Third, to regulate the amount of food eaten daily, all mice in the current study were housed separately. However, long-term isolation induces chronic stress and an increase in thermogenesis needs, which leads to an increase in energy expenditure [59]. To avoid the confounding effect of isolation housing on our ABA protocol, future study will need to be conducted under conditions in which two animals are housed in a large cage separated with a wire mesh divider to permit sensory interaction but not physical contact that disrupts individual dietary adjustment. Fourth, our study continued the FR session for 3.5 weeks (from days 11 to 35), which mimics the clinical fact that AN is a disease with a chronic course. However, Méquinion et al. [12] showed that ABA-loaded mice with time-restricted feeding display a lower wheel running activity after 35th day than AL group, which is concomitant with the slight but significant increase in energy expenditure in their ABA group when compared to the FR group. Given that the previous findings suggest that installation of a real chronic phase requires at least 5 weeks, future study will be conducted with an extended experimental duration to validate the "chronic" phase in our modified ABA protocol. Lastly, we analyzed the total adiponectin level in plasma using a commercial ELISA kit. However, blood adiponectin has three forms that differ from each other in multimerization and biological activity: low molecular weight (LMW; trimer), medium molecular weight (MMW; hexamer), high molecular weight (HMW; dodecamer and 18-mer), and total adiponectin [38]. HMW adiponectin appears to be the most active form, and the ratio of HMW adiponectin to total adiponectin is closely correlated with insulin sensitivity [60, 61]. Concerning the relation between adiponectin isoforms and AN, the findings of previous studies are inconsistent: the HMW adiponectin level of AN patients was higher than that of controls [62], a low ratio of HMW to total adiponectin and a high ratio of LMW to total adiponectin in AN were found [63], and the ratio of HMW to total adiponectin in AN was higher than that in controls [64]. This inconsistency is partly due to differences in patient characteristics, such as age, period of illness, or history of treatment. Thus, future study will be needed to analyze the effect of our ABA paradigm on blood adiponectin isoforms to clarify the relation between the pathophysiology of AN and adiponectin.

In conclusion, the present study found that a chronic food restriction-related elevation in the plasma adiponectin level was dampened in adolescent female ABA mice, which suggests that our method may be appropriate for developing a model of adolescent female AN. The plasma adiponectin level may be a useful candidate biomarker for the status or stage of AN.

## Author Contributions

**Conceptualization:** Toru Kuriyama, Yusuke Murata, Kenji Ohe.

**Data curation:** Toru Kuriyama, Yusuke Murata, Reika Ohtani, Rei Yahara, Soichiro Nakashima, Masayoshi Mori.

**Formal analysis:** Toru Kuriyama, Yusuke Murata, Masayoshi Mori.

**Investigation:** Toru Kuriyama, Yusuke Murata, Reika Ohtani, Rei Yahara, Soichiro Nakashima.

**Methodology:** Toru Kuriyama.

**Project administration:** Yusuke Murata.

**Supervision:** Yusuke Murata, Kenji Ohe, Kazunori Mine, Munechika Enjoji.

**Validation:** Yusuke Murata.

**Writing – original draft:** Toru Kuriyama, Yusuke Murata.

**Writing – review & editing:** Yusuke Murata, Kenji Ohe, Kazunori Mine, Munechika Enjoji.

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
