## [Decision Letter · Decision Letter 0]

12 Jun 2023

PONE-D-23-09382Modified activity-based anorexia paradigm dampens chronic food restriction-induced hyperadiponectinemia in adolescent female micePLOS ONE

Dear Dr. Murat,

Thank you for submitting your manuscript to PLOS ONE. After careful consideration, we feel that it has merit but does not fully meet PLOS ONE’s publication criteria as it currently stands. Therefore, we invite you to submit a revised version of the manuscript that addresses the points raised during the review process.

#REVIEWER 1

The study proposed in the manuscript “Modified activity-based anorexia paradigm dampens chronic food restriction-induced hyperadiponectinemia in adolescent female mice”, as written in the introduction section, aimed “to validate a modified FRW model that uses female mice to mimic the symptomatic features of clinical AN, such as hyperactivity, amenorrhea, and spontaneous food restriction. The second aim was “to investigate the effects of acute and chronic food restriction on the plasma adiponectin levels of experimental groups based on the presence or absence of wheel running and target weight loss”. To do so, authors conducted a study on young female mice separated and randomly assigned to one of the four experimental groups, ad libitum, ad libitum and wheel running (WR), food restriction (FR), food restriction and wheel running (ABA), during 24 days. For animal submitted to food restriction, these duration was divided in a first week of “acute phase” and a “chronic phase”. During the acute phase, animals were submitted to a 60% food restriction during 1 week leading to a 30% body weight loss, and to a 50% food restriction allowing to maintain this body weight for 4 weeks. Main significant results are i) an increase in light phase activity for ABA mice vs WR mice, ii) a disappearance or non-appearance of estrous cycle in FR and ABA mice, iii) an increase in plasma level of adiponectin in FR and ABA mice at the end point.

From these results the authors first conclude that ABA mice displayed “hyperactivity in the resting phase, amenorrhea, and a tendency to spontaneously restrict feeding “and considered that these characteristics features closely resemble the clinical symptoms of AN”. They finally also concluded that “the present study found that a chronic food restriction-related elevation in the plasma adiponectin level was dampened in adolescent female ABA mice, which validated our method as being an effective model of adolescent female AN. The plasma adiponectin level was shown to be a useful biomarker for AN.”

They also clearly discussed some limits of the study like the small sample size, the fact that only adiponectin was assessed while there are probably numerous endocrine changes, and finally the assessment of total adiponectin only while “HMW adiponectin appears to be the most active form, and the ratio of HMW adiponectin to total adiponectin is closely correlated with insulin sensitivity”.

This study provides novel results due to the very young age of the mice and the severity of the weight loss for this age. Data are fully available. The manuscript is presented in an intelligible fashion and written in standard English. However several changes and new experiments should improve the study and bring it up to the scientific standard of the journal.

Broad comments

About the novelty of the study:

Authors refer to two previous studies on ABA mouse model, Mequinion et al., 2015 and Frintrop et al., 2018. In Mequinion’s study, to validate the model, numerous parameters were assessed, like body composition, hourly follow-up of physical activity, energy expenditure, several plasma metabolites, glycaemia, or liver glycogen. Frintrop’s study, which animal protocol is close to that of the manuscript, assessed more or less the same parameters as the current study except adiponectin, and was conducted on rats. Compared to Frintrop’s study, the novelty of the current one consist in the transfer from rat to mouse and the assessment of total adiponectin. In a recent study, also cited by the authors, Tirelle et al. (2021) assessed plasma adiponectin in male and female mice of an ABA model. ABA female mice displayed a 25% body weight loss, plasma leptin level close to that of AL mice and a non-significant increase in total adiponectin. This last result should be more discussed in the manuscript. They also showed a high anticipatory activity in ABA female mice.

About the conclusion and the supporting data:

The authors conclude the manuscript by stating that i) the study validate the method as being an effective model of adolescent female AN, and ii) that plasma adiponectin level was shown to be a useful biomarker for AN. With regard to the data presented and the complexity of the pathology, this is at least an over-statement and must be modified.

About the experimental protocol:

In this study, female mice are only around 31 days old at the beginning of the FR/ABA protocol. At this age they only have little stored fat mass. So, the 60% FR seems to be very severe as the 30% body weight loss. Even if the study obtained the ethical approval of the University it is to note that it is usually admitted that the acceptable weight loss limit is 20% for adult. At this age, mice are supposed to grow fast. To better describe and understand the model and be able to try to do some comparisons with anorexia, it would have been useful to give some data on animal size for instance and also on body composition. Another point is the duration of the protocol. Mequinion et al. (2015) showed that ABA mice display a decrease in wheel running activity around day 35, while AL mice do not. Before and after this shift ABA mice also display different energy expenditure. These data suggest that the installation of a real chronic phase requires at least 5 weeks, which is the duration of the FR protocol in the current study. So, maybe one or two more weeks could give different information and allow to be sure to be in a chronic phase.

It is to note that all animals are housed separately. This is known to induce a chronic stress and an increase in thermogenesis needs, both of them leading to an increase in energy expenditure (Zgheib S, et al., PLoS ONE 2014, doi:10.1371/journal.pone.0103775).

The criteria leading to determine if there is or not an estrous cycle should be presented in the Materials and Methods section. Is it only a lack of diestrous, or the absence of most of/all the phases?

About the description of the model. The study aims to validate a modified ABA protocol as relevant model of anorexia nervosa. As mentioned in the comment on conclusion and supporting data, much more data, like basal glycaemia and glucose tolerance test or insulin sensitivity, bone status and body composition are necessary to describe and validate the modified ABA mice as a model of anorexia. 

One part of the study focus on self-starvation to try to reinforce the similarity with human pathological behavior. Differences are not significant here, but above all it is necessary to take into account the study of Boakes et al. (The role of drinking in the suppression of food intake by recent activity. Behav Neurosci 115: 718–730, 2001) that demonstrated that the use of hydrated food suppresses this behavior in ABA model and explained that running animals are thirsty and drink a lot before eating reaching a satiety sensation that decreases their food intake. This study showed that this criterion is not relevant to validate the model.

#REVIEWER 2

The study is interesting and sheds light over the importance of adiponectin in the pathogenesis of eating disorders. I would like to ask the authors to better explain why they have modified the ABA model. I think the fact that this has been done along with adiponectin level checking is somehow confusing

We look forward to receiving your revised manuscript.

Kind regards,

Fabio Vasconcellos Comim

Academic Editor

PLOS ONE

Reviewers' comments:

Reviewer's Responses to Questions

**Comments to the Author**

1. Is the manuscript technically sound, and do the data support the conclusions?

Reviewer #1: Partly

Reviewer #2: Yes

2. Has the statistical analysis been performed appropriately and rigorously? 

Reviewer #1: I Don't Know

Reviewer #2: Yes

3. Have the authors made all data underlying the findings in their manuscript fully available?

Reviewer #1: Yes

Reviewer #2: No

4. Is the manuscript presented in an intelligible fashion and written in standard English?

Reviewer #1: Yes

Reviewer #2: Yes

5. Review Comments to the Author

Reviewer #1: The study proposed in the manuscript “Modified activity-based anorexia paradigm dampens chronic food restriction-induced hyperadiponectinemia in adolescent female mice”, as written in the introduction section, aimed “to validate a modified FRW model that uses female mice to mimic the symptomatic features of clinical AN, such as hyperactivity, amenorrhea, and spontaneous food restriction. The second aim was “to investigate the effects of acute and chronic food restriction on the plasma adiponectin levels of experimental groups based on the presence or absence of wheel running and target weight loss”. To do so, authors conducted a study on young female mice separated and randomly assigned to one of the four experimental groups, ad libitum, ad libitum and wheel running (WR), food restriction (FR), food restriction and wheel running (ABA), during 24 days. For animal submitted to food restriction, these duration was divided in a first week of “acute phase” and a “chronic phase”. During the acute phase, animals were submitted to a 60% food restriction during 1 week leading to a 30% body weight loss, and to a 50% food restriction allowing to maintain this body weight for 4 weeks. Main significant results are i) an increase in light phase activity for ABA mice vs WR mice, ii) a disappearance or non-appearance of estrous cycle in FR and ABA mice, iii) an increase in plasma level of adiponectin in FR and ABA mice at the end point.

From these results the authors first conclude that ABA mice displayed “hyperactivity in the resting phase, amenorrhea, and a tendency to spontaneously restrict feeding “and considered that these characteristics features closely resemble the clinical symptoms of AN”. They finally also concluded that “the present study found that a chronic food restriction-related elevation in the plasma adiponectin level was dampened in adolescent female ABA mice, which validated our method as being an effective model of adolescent female AN. The plasma adiponectin level was shown to be a useful biomarker for AN.”

They also clearly discussed some limits of the study like the small sample size, the fact that only adiponectin was assessed while there are probably numerous endocrine changes, and finally the assessment of total adiponectin only while “HMW adiponectin appears to be the most active form, and the ratio of HMW adiponectin to total adiponectin is closely correlated with insulin sensitivity”.

This study provides novel results due to the very young age of the mice and the severity of the weight loss for this age. Data are fully available. The manuscript is presented in an intelligible fashion and written in standard English. However several changes and new experiments should improve the study and bring it up to the scientific standard of the journal.

Broad comments

About the novelty of the study:

Authors refer to two previous studies on ABA mouse model, Mequinion et al., 2015 and Frintrop et al., 2018. In Mequinion’s study, to validate the model, numerous parameters were assessed, like body composition, hourly follow-up of physical activity, energy expenditure, several plasma metabolites, glycaemia, or liver glycogen. Frintrop’s study, which animal protocol is close to that of the manuscript, assessed more or less the same parameters as the current study except adiponectin, and was conducted on rats. Compared to Frintrop’s study, the novelty of the current one consist in the transfer from rat to mouse and the assessment of total adiponectin. In a recent study, also cited by the authors, Tirelle et al. (2021) assessed plasma adiponectin in male and female mice of an ABA model. ABA female mice displayed a 25% body weight loss, plasma leptin level close to that of AL mice and a non-significant increase in total adiponectin. This last result should be more discussed in the manuscript. They also showed a high anticipatory activity in ABA female mice.

About the conclusion and the supporting data:

The authors conclude the manuscript by stating that i) the study validate the method as being an effective model of adolescent female AN, and ii) that plasma adiponectin level was shown to be a useful biomarker for AN. With regard to the data presented and the complexity of the pathology, this is at least an over-statement and must be modified.

About the experimental protocol:

In this study, female mice are only around 31 days old at the beginning of the FR/ABA protocol. At this age they only have little stored fat mass. So, the 60% FR seems to be very severe as the 30% body weight loss. Even if the study obtained the ethical approval of the University it is to note that it is usually admitted that the acceptable weight loss limit is 20% for adult. At this age, mice are supposed to grow fast. To better describe and understand the model and be able to try to do some comparisons with anorexia, it would have been useful to give some data on animal size for instance and also on body composition. Another point is the duration of the protocol. Mequinion et al. (2015) showed that ABA mice display a decrease in wheel running activity around day 35, while AL mice do not. Before and after this shift ABA mice also display different energy expenditure. These data suggest that the installation of a real chronic phase requires at least 5 weeks, which is the duration of the FR protocol in the current study. So, maybe one or two more weeks could give different information and allow to be sure to be in a chronic phase.

It is to note that all animals are housed separately. This is known to induce a chronic stress and an increase in thermogenesis needs, both of them leading to an increase in energy expenditure (Zgheib S, et al., PLoS ONE 2014, doi:10.1371/journal.pone.0103775).

The criteria leading to determine if there is or not an estrous cycle should be presented in the Materials and Methods section. Is it only a lack of diestrous, or the absence of most of/all the phases?

About the description of the model. The study aims to validate a modified ABA protocol as relevant model of anorexia nervosa. As mentioned in the comment on conclusion and supporting data, much more data, like basal glycaemia and glucose tolerance test or insulin sensitivity, bone status and body composition are necessary to describe and validate the modified ABA mice as a model of anorexia.

One part of the study focus on self-starvation to try to reinforce the similarity with human pathological behavior. Differences are not significant here, but above all it is necessary to take into account the study of Boakes et al. (The role of drinking in the suppression of food intake by recent activity. Behav Neurosci 115: 718–730, 2001) that demonstrated that the use of hydrated food suppresses this behavior in ABA model and explained that running animals are thirsty and drink a lot before eating reaching a satiety sensation that decreases their food intake. This study showed that this criterion is not relevant to validate the model.

Reviewer #2: The study is interesting and sheds light over the importance of adiponectin in the pathogenesis of eating disorders. I would like to ask the authors to better explain why they have modified the ABA model. I think the fact that this has been done along with adiponectin level checking is somehow confusing

---

## [Author Response · Author response to Decision Letter 0]

25 Jun 2023

[Response to reviewer 1]

1) About the novelty of the study:

Authors refer to two previous studies on ABA mouse model, Mequinion et al., 2015 and Frintrop et al., 2018. In Mequinion’s study, to validate the model, numerous parameters were assessed, like body composition, hourly follow-up of physical activity, energy expenditure, several plasma metabolites, glycaemia, or liver glycogen. Frintrop’s study, which animal protocol is close to that of the manuscript, assessed more or less the same parameters as the current study except adiponectin, and was conducted on rats. Compared to Frintrop’s study, the novelty of the current one consist in the transfer from rat to mouse and the assessment of total adiponectin. In a recent study, also cited by the authors, Tirelle et al. (2021) assessed plasma adiponectin in male and female mice of an ABA model. ABA female mice displayed a 25% body weight loss, plasma leptin level close to that of AL mice and a non-significant increase in total adiponectin. This last result should be more discussed in the manuscript. They also showed a high anticipatory activity in ABA female mice.

We appreciate your helpful comment. To describe an intensive consideration concerning the data of Tirelle et al. (2021), the sentences, “Little is known about the impact of the ABA paradigm on the blood adiponectin level. Tirelle et al. [49] found no change in the blood adiponectin levels of intact and ABA paradigm-loaded mice. This study used young-adult mice and adopted a standard ABA protocol with time-restricted food access and wheel running [49] in contrast to our modified protocol with amount-restricted food access to achieve a target weight reduction. The present study is the first to demonstrate an effect from this protocol on the plasma adiponectin level of adolescent female mice.” were corrected as follows (page 25, line 433 to page 26, line 457):

“Although little is known about the impact of the ABA paradigm on the blood adiponectin level, Tirelle et al. [50] found no change in the blood adiponectin levels of intact and ABA paradigm-loaded female mice. That study also demonstrated that the plasma leptin concentration of their AL group was comparable to that of their ABA group, which displayed a 25% body weight loss. Because leptin is secreted from adipose tissue, the blood leptin level in ABA-loaded animals has been shown to be reduced by the food restriction-induced decrement of fat mass [12, 51, 52]. It is difficult to understand how there could be a decrease in lean mass and an increase in fat mass in female mice loaded with ABA paradigm, however, it may explain why no alteration in plasma leptin level was found between in AL and ABA group [50]. Despite adiponectin being secreted from adipocytes, the blood adiponectin level has been negatively correlated with body fat mass [17]. Thus, the increased adiposity in female ABA mice of the Tirelle study may have led to the decrease in adiponectin, which should have been high under the poor nutrition state [50]. Although we did not examine fat mass, post-decapitation dissection confirmed that little fatty tissue remained. Our modified protocol with amount-restricted food access is the first study to demonstrate an effect on the plasma adiponectin level of adolescent female mice of achieving a target weight reduction. Of interest, the male ABA mice of the Tirelle study showed higher decreases in fat mass and plasma leptin level than AL mice, but not plasma adiponectin level [50]. Because male and female mice exhibit different responses to the ABA model [53], the ABA-induced increment in adiponectin may be specific to female mice. Indeed, an elevated level of adiponectin has been associated with reduced basal gonadotropin-releasing hormone and luteinizing hormone in mice [54], which results in FHA. Future studies will be needed to clarify whether or not the ABA-induced change in adiponectin is sex-specific.”

Four new references that were included in the above were added (#51, Pardo et al., 2010; #52, Gelegen et al., 2007; #53, Achamrah et al., 2017; #54, Dobrzyn et al., 2018) (page 33, line 644 to page 34, line 653).

2) About the conclusion and the supporting data:

The authors conclude the manuscript by stating that i) the study validate the method as being an effective model of adolescent female AN, and ii) that plasma adiponectin level was shown to be a useful biomarker for AN. With regard to the data presented and the complexity of the pathology, this is at least an over-statement and must be modified. 

To address this problem, the sentence, “These results indicate that the plasma adiponectin level would be a useful biomarker for AN.” was corrected as follows (page 3, lines 37 to 38):

“These results indicate that the plasma adiponectin level may be a useful candidate biomarker for the status or stage of AN.”

Also, the sentence, “In conclusion, the present study found that a chronic food restriction-related elevation in the plasma adiponectin level was dampened in adolescent female ABA mice, which validated our method as being an effective model of adolescent female AN. The plasma adiponectin level was shown to be a useful biomarker for AN.” was corrected as follows (page 30, lines 514 to 518):

“In conclusion, the present study found that a chronic food restriction-related elevation in the plasma adiponectin level was dampened in adolescent female ABA mice, which suggests that our method may be appropriate for developing a model of adolescent female AN. The plasma adiponectin level may be a useful candidate biomarker for the status or stage of AN.

3) About the experimental protocol:

 3-1) In this study, female mice are only around 31 days old at the beginning of the FR/ABA protocol. At this age they only have little stored fat mass. So, the 60% FR seems to be very severe as the 30% body weight loss. Even if the study obtained the ethical approval of the University it is to note that it is usually admitted that the acceptable weight loss limit is 20% for adult. At this age, mice are supposed to grow fast. To better describe and understand the model and be able to try to do some comparisons with anorexia, it would have been useful to give some data on animal size for instance and also on body composition.

I am grateful for your suggestion. According to your comment, the following sentence for ethical concern was added (page 8, lines 126 to 129):

“The current study obtained approval of the University Ethics Committee, even though the 60% FR necessary to achieve a 30% body weight loss seems to be very severe when the acceptable weight loss limit is usually considered 20% for adult mice.”

The following sentences were added as a limitation of the study (page 23, lines 390 to 395):

“Fourth, we evaluated symptomatic features of AN, such as hyperactivity, amenorrhea, and spontaneous food restriction. However, other characteristics of AN that many studies have assessed; including basal glycaemia, glucose tolerance, bone status, and body composition; were not examined. Future study that evaluates physiological abnormalities will need to be conducted for comparison with previous ABA paradigms, which will emphasize the uniqueness of our model.”

 3-2) Another point is the duration of the protocol. Mequinion et al. (2015) showed that ABA mice display a decrease in wheel running activity around day 35, while AL mice do not. Before and after this shift ABA mice also display different energy expenditure. These data suggest that the installation of a real chronic phase requires at least 5 weeks, which is the duration of the FR protocol in the current study. So, maybe one or two more weeks could give different information and allow to be sure to be in a chronic phase.

 According to your comment, the following sentences on the real “chronic phase” were added (page 28, line 491 to page 29, line 499):

“Fourth, our study continued the FR session for 3.5 weeks (from days 11 to 35), which mimics the clinical fact that AN is a disease with a chronic course. However, Méquinion et al. [12] showed that ABA-loaded mice with time-restricted feeding display a lower wheel running activity after 35th day than AL group, which is concomitant with the slight but significant increase in energy expenditure in their ABA group when compared to the FR group. Given that the previous findings suggest that installation of a real chronic phase requires at least 5 weeks, future study will be conducted with an extended experimental duration to validate the “chronic” phase in our modified ABA protocol.”

 3-3) It is to note that all animals are housed separately. This is known to induce a chronic stress and an increase in thermogenesis needs, both of them leading to an increase in energy expenditure (Zgheib S, et al., PLoS ONE 2014, doi:10.1371/journal.pone.0103775).

 According to your suggestion, the following sentences were added as an additional limitation (page 28, lines 485 to 491):

“Third, to regulate the amount of food eaten daily, all mice in the current study were housed separately. However, long-term isolation induces chronic stress and an increase in thermogenesis needs, which leads to an increase in energy expenditure [59]. To avoid the confounding effect of isolation housing on our ABA protocol, future study will need to be conducted under conditions in which two animals are housed in a large cage separated with a wire mesh divider to permit sensory interaction but not physical contact that disrupts individual dietary adjustment.”

A supporting reference (#59, Zgheib et al., 2014) was added (page 34, lines 665 to 667).

 3-4) The criteria leading to determine if there is or not an estrous cycle should be presented in the Materials and Methods section. Is it only a lack of diestrous, or the absence of most of/all the phases?

 I apologize that it is so confusing. The sentence, “Because the estrous cycle of a healthy mouse takes 4 days [22], the incidence of the estrous cycle was measured in 4-day blocks.” was corrected as follows (page 9, lines 150 to 154):

“Because the estrous cycle of a healthy mouse takes 4 days [22], the incidence of the estrous cycle was determined by vaginal smear cell morphology in 4-day blocks that met any of the following conditions: transition from proestrous to estrous, estrous, or transition from estrous to proestrous.”

3-5) About the description of the model. The study aims to validate a modified ABA protocol as relevant model of anorexia nervosa. As mentioned in the comment on conclusion and supporting data, much more data, like basal glycaemia and glucose tolerance test or insulin sensitivity, bone status and body composition are necessary to describe and validate the modified ABA mice as a model of anorexia.

Similar to our response #3-1, sentences about the necessity for additional data were added (page 23, lines 390 to 395). 

3-6) One part of the study focus on self-starvation to try to reinforce the similarity with human pathological behavior. Differences are not significant here, but above all it is necessary to take into account the study of Boakes et al. (The role of drinking in the suppression of food intake by recent activity. Behav Neurosci 115: 718–730, 2001) that demonstrated that the use of hydrated food suppresses this behavior in ABA model and explained that running animals are thirsty and drink a lot before eating reaching a satiety sensation that decreases their food intake. This study showed that this criterion is not relevant to validate the model.

 I am grateful for your comment. Accordingly, the sentences, “Méquinion et al. [18] fed their mice a constant amount of food throughout the experimental period, whereas we adopted a protocol that is able to fine-tune the amount of food given in order to maintain a target weight. Because it is difficult to compare the results of studies with different methodologies, further studies will be necessary after matching the experimental conditions.” were corrected as follows (page 24, lines 406 to 414):

“Méquinion et al. [12] fed their mice a constant amount of food throughout the experimental period, whereas we adopted a protocol that is able to fine-tune the amount of food given in order to maintain a target weight, thus it is difficult to compare the results of these studies because of the different methodologies. Of interest, because running animals were thirsty and drank a lot before reaching a satiety sensation which decreases their food intake, Boakes and Juraskova [36] demonstrated that use of wet and mashed food suppresses drinking-induced decrement of feeding behavior in an ABA model. Further studies will be necessary after optimizing the experimental conditions.”

A reference used in the above was added (#36, Boakes and Juraskova, 2001) (page 32, lines 603 to 604).

[Response to reviewer 2]

The study is interesting and sheds light over the importance of adiponectin in the pathogenesis of eating disorders. I would like to ask the authors to better explain why they have modified the ABA model. I think the fact that this has been done along with adiponectin level checking is somehow confusing

I appreciate your comment and apologize that the purpose in this study is so confusing. To better understand our aim, the sentences and paragraph were rearranged as follows:

1) The paragraph about the first aim, “When developing experimental animal models of AN, … such as hyperactivity, amenorrhea, and spontaneous food restriction.” was moved to page 3, line 52 in the revised version. 

Accordingly, the sentence, “Thus, valid animal models of AN must be prepared to clarify the pathogenesis and develop therapeutic agents for AN.” was added to the revised paper (page 3, lines 49 to 51). 

The sentence, “Only female mice were used because of a higher incidence of AN in women and girls than in men and boys (sex ratios of approximately to 10:1 to 15:1 [7]).” was moved to page 5, line 73.

 2) The paragraph about relation between adiponectin and AN, “AN is associated with multiple, profound endocrine disturbances …but also in healthy underweight young people (BMI < 18.5) [13].” was moved to page 5, line 76.

 The sentence, “we developed a modified mouse model for this study that would allow us to determine if circulating adiponectin can be used as a specific biomarker that reflects the disease state of AN and to elucidate how the effect of weight loss on the blood adiponectin level differs between 1) weight control intervention (dietary restriction and experimental AN) and 2) the acute and chronic phases of AN.” was slightly modified and moved to page 6, line 90, with the term “In light of the above,” at the beginning.

---

## [Decision Letter · Decision Letter 1]

10 Jul 2023

Modified activity-based anorexia paradigm dampens chronic food restriction-induced hyperadiponectinemia in adolescent female mice

PONE-D-23-09382R1

Dear Dr. Murata,

We’re pleased to inform you that your manuscript has been judged scientifically suitable for publication and will be formally accepted for publication once it meets all outstanding technical requirements.

Kind regards,

Fabio Vasconcellos Comim

Academic Editor

PLOS ONE

Additional Editor Comments (optional):

Dear Dr Murata,

I am glad to inform that after a careful review, your manuscript was accepted for publication in PLOS ONE.

Congratulations.

Kind regards,

Academic editor

Reviewers' comments:

Reviewer's Responses to Questions

**Comments to the Author**

1. If the authors have adequately addressed your comments raised in a previous round of review and you feel that this manuscript is now acceptable for publication, you may indicate that here to bypass the “Comments to the Author” section, enter your conflict of interest statement in the “Confidential to Editor” section, and submit your "Accept" recommendation.

Reviewer #1: All comments have been addressed

Reviewer #2: (No Response)

2. Is the manuscript technically sound, and do the data support the conclusions?

Reviewer #1: Yes

Reviewer #2: (No Response)

3. Has the statistical analysis been performed appropriately and rigorously? 

Reviewer #1: Yes

Reviewer #2: (No Response)

4. Have the authors made all data underlying the findings in their manuscript fully available?

Reviewer #1: Yes

Reviewer #2: (No Response)

5. Is the manuscript presented in an intelligible fashion and written in standard English?

Reviewer #1: Yes

Reviewer #2: (No Response)

6. Review Comments to the Author

Reviewer #1: (No Response)

Reviewer #2: The manuscript has improved a lot. I have no further modifications. Please see the comments for editors section for further details

7. PLOS authors have the option to publish the peer review history of their article (what does this mean?). If published, this will include your full peer review and any attached files.

Reviewer #1: **Yes: **Christophe Chauveau

Reviewer #2: **Yes: **Rami Bou Khalil

---

## [Editor Report · Acceptance letter]

13 Jul 2023

PONE-D-23-09382R1 

Modified activity-based anorexia paradigm dampens chronic food restriction-induced hyperadiponectinemia in adolescent female mice 

Dear Dr. Murata:

I'm pleased to inform you that your manuscript has been deemed suitable for publication in PLOS ONE. Congratulations! Your manuscript is now with our production department. 

Kind regards, 

on behalf of

Prof Fabio Vasconcellos Comim 

Academic Editor

PLOS ONE